# Combustible cigarettes, heated tobacco products, combined product use, and periodontal disease: A cross-sectional JASTIS study

Takashi Yoshioka[1], Takahiro Tabuchi[2]*

1 Center for Innovative Research for Communities and Clinical Excellence (CiRC2LE), Fukushima Medical University, Fukushima-shi, Fukushima, Japan, 2 Cancer Control Center, Osaka International Cancer Institute, Otemae, Chuo-ku, Osaka-shi, Osaka, Japan

* tabuchitak@gmail.com

## Abstract

### Background

Combustible cigarettes have detrimental effects on periodontal disease. However, little evidence is available regarding new heated tobacco product (HTP) use and combined product use (both combustible cigarettes and HTPs). This study aimed to examine the association of combustible cigarettes, HTPs, and combined product use with periodontal disease simultaneously.

### Materials and methods

This cross-sectional study was conducted using data from the 2019 arm of the longitudinal Japan "Society and New Tobacco" Internet Survey. Combustible cigarette users, HTP users, combined product users, never-users, and former users' data were separately obtained. In the present study, the primary outcome was self-reported periodontal disease. We estimated adjusted prevalence ratios (PRs) and confidence intervals (CIs) using multivariable modified Poisson regression analysis after adjusting for 12 confounders.

### Results

Of the 10,439 JASTIS respondents, the numbers of users of combustible cigarettes only, HTPs only, and both products were 1,304, 437, and 1,049, respectively. Compared with never-users, HTP use was significantly associated with the prevalence of self-reported periodontal diseases (PR 1.43, 95% CI 1.03–1.62). Moreover, former users, combustible cigarette users, and combined product users also showed significant associations (PR 1.56, 95% CI 1.35–1.80; PR 1.29, 95% CI 1.03–1.62; and PR 1.55, 95% CI 1.20–1.99, respectively).

**Data Availability Statement:** The data used in this study are not available in a public repository because they contain personally identifiable or potentially sensitive participants' information.

Based on the regulations for ethical guidelines in Japan, the Research Ethics Committee of the Osaka International Cancer Institute has imposed restrictions on the dissemination of the data collected in this study. All data inquiries will be channeled through Tabuchi (tabuchitak@gmail.com) to Osaka Cancer Institute Institutional Ethics Committee.

**Funding:** TT was supported by Health Labour Sciences Research Grants (20FA1005; 19FA0501; 19FA2001; and 19FA1011; https://mhlw-grants.niph.go.jp/), and Japan Society for the Promotion of Science (JSPS) KAKENHI Grants (18H03062; https://www.jsps.go.jp/english/e-grants/). The funders had no role in study design, data collection and analysis, decision to publish, or preparation of the manuscript.

**Competing interests:** The authors have declared that no competing interests exist.

## Conclusions

Users of HTPs, combustible cigarettes, and combined products as well as former users were all significantly associated with a higher prevalence of periodontal diseases compared to never-users.

## Introduction

Smoking is a global public health problem, as it is a leading risk factor for cancer and chronic respiratory diseases, and was the second leading risk factor for early death and disability worldwide in 2015 [1]. Because of the heavy disease burden due to smoking, tobacco control has been an important issue among healthcare professionals and policymakers [1, 2].

Recently, heated tobacco products (HTPs, also known as "heat-not-burn" tobacco devices) have appeared as a new way to "smoke" tobacco [3]. The first of the new products, IQOS, was launched in Italy and Japan, and the markets have been growing globally [4, 5]. Tobacco industries advertise that HTPs are less harmful to health than combustible cigarettes, presenting HTPs as appealing alternatives to use for smoking cessation [3, 5, 6]. However, recent studies indicate that HTPs are not always safer than combustible cigarettes. It has been reported that HTPs contain more harmful substances than combustible cigarettes [3], and their use has cytotoxic effects on human bronchial epithelium [7]. As HTPs were launched within six years of their invention, nobody knows all their potentially harmful and long-term effects. Therefore, global investigations are needed to clarify the health impacts of HTPs.

Periodontal disease, which is characterized by the loss of periodontal ligament tissue and the collapse of surrounding alveolar bone tissue, is prevalent (7.4%) globally [8]. Periodontal disease causes systemic inflammation and is associated with many systemic diseases, such as cardiovascular diseases, cancers, rheumatoid arthritis, and chronic kidney diseases [9]. Periodontal disease, especially severe chronic periodontal disease (SCP), is the second leading cause of disability-adjusted life-years (DALYs) associated with oral conditions [8]. The estimated global burden of SCP accounted for 3.5 million DALYs (95% uncertainty intervals [UI]: 1.4 to 7.2) with an increasing trend from 1995 to 2015 [8]. Taken together, it is clear that reducing the incidence and the prevalence of periodontal disease is an important public health issue, as reducing the prevalence of the condition may lead to reductions in both the associated systemic diseases and subsequent economic burdens.

Smoking is known to have detrimental effects on periodontal health because it changes periodontal microorganisms in the mouth of a smoker, causing periodontal tissue breakdown [10]. The global smoking-attributed burden of periodontal disease was also high, which accounted for 251,160 DALYs (95% UI: 190,721 to 324,241) among 186 countries in 2015 [11]. Recent studies show that electronic cigarettes (e-cigarettes) and combustible cigarettes are associated with these harmful effects, leading to periodontal disease [12]. However, little evidence is available concerning the relationship between HTPs and periodontal diseases. Furthermore, it remains unclear whether such relationships change depending on the type of tobacco product used, such as combustible cigarettes or HTPs. Therefore, we aimed to examine the association of smoking—according to the types of tobacco products (i.e., combustible cigarettes or HTPs) and combined product use—with the prevalence of periodontal disease using data from a large population sample.

## Methods

### Study design and setting

For this cross-sectional study, we used data from the 2019 Japanese "Society and New Tobacco" Internet Survey (JASTIS). The JASTIS longitudinal cohort study used a series of internet surveys, from 2015 to 2019, to collect information about the prevalence of new tobacco product use, including HTPs and e-cigarettes, information about the use of conventional tobacco products, and participants' demographic and socioeconomic data. The resulting data were made available for use by other researchers and collaborative research projects. The 2019 survey data were collected between the 2nd and 28th of February 2019. The online questionnaire was designed such that respondents had to answer each question before they were able to proceed to the next, ensuring all questions were answered. Participants of the JASTIS were recruited via a survey panel provided from a major internet research agency in Japan (Rakuten Insight). The agency keeps about 2.3 million panelists and their socioeconomic status, such as education levels, household income, and marital status. The survey panel comprised those who were recruited initially via services of the Rakuten agency group. For the 2015–2018 baseline survey, participants were randomly sampled from the total panelists from the Rakuten Insight database. The follow-up surveys in other years were conducted from all respondents who previously participated in JASTIS. In the 2019 survey, 9,262 participants were recruited from the follow-up survey. The response rate was 62.5% (9,262 /14,825). In addition to the follow-up survey, the 2019 survey collected new participants (1,738) aged 15–24 from the panel because of the small population compared with other age groups. The additional survey was closed when the target number of respondents who had answered the questionnaire was met. In total, 11,000 respondents participated in JASTIS 2019. Further detailed information regarding the JASTIS is described in the study profile [13].

### Inclusion and exclusion criteria

This study included data from all respondents of the 2019 survey except those whose responses were inconsistent with the information they had provided in the earlier surveys (2015 to 2018). On the other hand, this study excluded those whose responses were straight-lining or contained discrepancies. For example, we excluded surveys as straight-lining responses if the respondents chose the same answer number for all questions in a set of questions. We also excluded respondents as responses with discrepancies if they reported an amount of tobacco product use but had indicated that they had never used, or were only former users of, tobacco products. In addition to these exclusion criteria, we performed an attention check for respondents, using the question: "Please choose the second from the bottom." Using this attention check, we excluded respondents who selected responses except the second answer from the bottom.

### Measurement of exposure (tobacco product use)

To evaluate the association between past tobacco product use, combustible tobacco use, HTP use, and combined use compared to non-smokers, this study defined the exposures to discriminate each smoking status clearly. Respondents defined as current tobacco product users were asked, "During the past 30 days, have you used each tobacco product? (paper-wrapped cigarettes, roll-your-own cigarettes, nicotine e-cigarettes, non-nicotine e-cigarettes, e-cigarettes with unknown nicotine content, Ploom tech, Ploom tech plus, Ploom S, IQOS, glo, cigars, pipes, chewing tobacco, snuff, and hookahs)"; response options were "Yes" and "No." Among the current users, types of products were classified as follows: combustible cigarette use

(paper-wrapped and roll-your-own cigarettes), HTP use (Ploom tech, Ploom tech plus, Ploom S, IQOS, and glo). We discriminated between the respondents who used only combustible cigarettes, HTP only users, and users of both products. Among non-current users, we separated former users from those who had never used tobacco products, which we determined from the survey data, as shown in S1 Table. Hence, each of the respondents included in our study was classified as one of the following: "never user," "former user," "combustible cigarette user," "HTP user," or "combined product user."

## Main outcome measures

We defined self-reported periodontal disease as a primary outcome. The JASTIS survey included the question, "Do you have any of the following chronic diseases (for which you have received a diagnosis or are regularly visiting the hospital): hypertension, diabetes, asthma, bronchitis or pneumonia, periodontal diseases, atopic dermatitis, otitis media, heart diseases, stroke, COPD, cancer, depression, or other mental disorders?" The response options were "not affected," "currently affected and regularly visiting a hospital (clinic)," and "currently affected and without regular (hospital/clinic) visits." The latter two responses about periodontal diseases were combined and defined as "currently affected by periodontal disease (overall periodontal disease)," and the latter one response was defined as "currently affected by periodontal disease under treatment (periodontal disease under treatment)." This questionnaire consists of the same questions and choices as those in the Comprehensive Survey of Living Conditions in Japan [14].

## Covariates

Based on the previous studies regarding the association between combustible cigarettes/e-cigarettes and periodontal disease, we selected age, sex, body mass index (BMI), socioeconomic status (marital status, educational levels, household income), alcohol use [12, 15, 16], routine dental checkups [12, 15], secondhand exposure of combustible cigarettes from others, secondhand aerosols of heated tobacco products from others [12], smoking pack-years [17], and comorbidities (hypertension, diabetes mellitus, heart disease) [17–19] as confounders. In addition to such known confounders, we further selected additional potential confounders (stroke, other tobacco product use) for constructing multivariable models. In total, we selected 12 confounders as covariates. These variables were categorized as follows: age (18–24 years, 25–34, 35–44, 45–54, 55–64, and 65–74), sex (male and female), body mass index, as kg/m$^2$ ($\leq$ 18.4, 18.5–24.9, 25–29.9, $\geq$ 30), alcohol use during the past 30 days, income ($\leq$ 9,999 US dollars/year, 10,000–39,999, 40,000–79,999, 80,000–119,999, $\geq$ 120,000, and "unknown"; calculated as 100 JP yen = 1 US dollar), education level (less than high school, high school graduate, college or associates degree, bachelor's degree, and master's or doctoral degree), Routine dental checkup, use of other tobacco products, secondhand smoke exposure from others (exposure to smoke/aerosols from combustible cigarettes and/or e-cigarettes or HTPs by others), comorbidities (hypertension, diabetes mellitus, bronchitis or pneumonia, heart diseases, and stroke), and smoking history calculated by pack-years ($\leq$ 5, 6–10, 11–20, 21–30, 31–40, 41–50, $\geq$ 51) [20].

## Statistical analysis

First, we calculated the number and frequency of each variable for use as baseline characteristics. Second, multivariable modified Poisson regression analysis was performed to estimate the prevalence ratio (PR) and confidence interval (CI) for the prevalence of self-reported periodontal disease as the primary analysis, adjusting for the 12 covariates [21, 22]. Additionally,

we performed two sensitivity analyses. In the first sensitivity analysis, we changed the definition of exposure from overall periodontal diseases to those under treatment. Furthermore, to validate the confounding selection in the main analysis, we performed a second sensitivity analysis which describes a directed acyclic graph (DAG), and constructed multivariable models based on confounders from the DAG (*i.e.* age, sex, educational level, routine dental checkup, secondhand exposure to combustible cigarettes from others, secondhand exposure to heated tobacco products or e-cigarettes from others, and smoking pack-years), and confirmed the consistencies of the results. To consider the multicollinearity of variables in the multivariable models, we calculated variance inflation factors (VIF) in both the main and sensitivity analyses. The data were analyzed using STATA version 15.1 (Stata Corp., College Station, TX, USA).

## Ethical considerations

We obtained web-based informed consent from all the respondents whose data from the JAS-TIS study were used in our research. This study was approved by the Institutional Review Board of Osaka International Cancer Institute (No. 1412175183) and the National Institute of Public Health (NIPH-IBRA#12112). The data were anonymized before statistical analyses.

## Results

Of 11,000 respondents, 10,439 subjects were included, as shown in Fig 1. The number of current tobacco product users was 2,790; of the current users, 1,304 were combustible cigarette users, 437 were HTP users, and 1,049 were combined product users, respectively. The overall prevalence of periodontal disease was 12.3% (1,279/10,439); periodontal prevalence for combustible tobacco product users, HTP users, and combined product users was 15.3% (200/1,304), 15.1% (66/437), and 19.4% (204/1,049), respectively. In contrast, the prevalence of periodontal disease for never-users was 8.0% (463/5,796). The baseline characteristics are shown in Table 1.

The results of multivariable modified Poisson regression analyses both in the main and sensitivity analyses were shown in Table 2. After adjustment for the 12 confounders, the PR of combustible products, HTPs, and combined use presented statistically significant associations (PR [95% CI]: 1.29 [1.03–1.62]; 1.43 [1.08–1.88]; and 1.55 [1.20–1.99]) compared with never-users, respectively. Moreover, former users were also significantly associated with the outcome (1.56 [1.35–1.80]). After changing the outcome from the overall prevalence of periodontal disease to that under treatment, similar significant associations were consistently found among former users, combustible tobacco product users, HTP users, and combined product users (PR [95% CI]: 1.85[1.48–2.30]; 1.77 [1.27–2.46]; 1.82 [1.21–2.74]; and 1.82 [1.26–2.63],

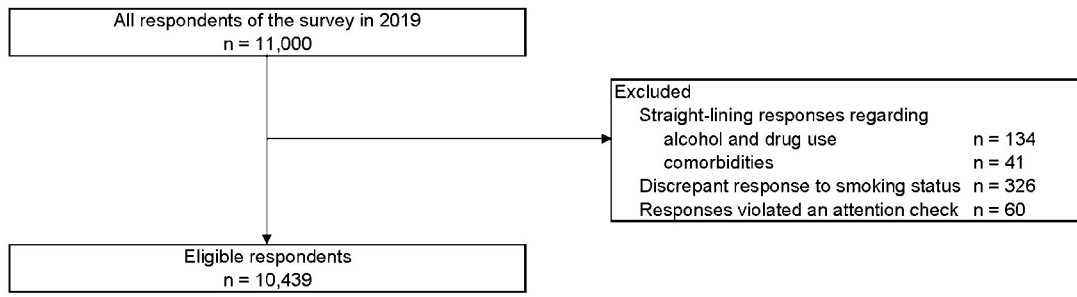

**Fig 1. Flow diagram of the study.**

**Table 1. Demographics of respondents.**

| | Never user | | Former user | | Current user N = 2,790 | | | | | | Total | |
| --- | --- | --- | --- | --- | --- | --- | --- | --- | --- | --- | --- | --- |
| | | | | | Only combustible cigarettes | | Only heated tobacco products | | Combined use | | | |
| | N = 5,796 | | N = 1,853 | | N = 1,304 | | N = 437 | | N = 1,049 | | N = 10,439 | |
| | n | % | n | % | n | % | n | % | n | % | n | % |
| **Age (years)** | | | | | | | | | | | | |
| 18–24 | 1,602 | 27.6 | 62 | 3.3 | 74 | 5.7 | 60 | 13.7 | 114 | 10.9 | 1,912 | 18.3 |
| 25–34 | 879 | 15.2 | 77 | 4.2 | 58 | 4.5 | 43 | 9.8 | 48 | 4.6 | 1,105 | 10.6 |
| 35–44 | 925 | 16.0 | 307 | 16.6 | 235 | 18.0 | 97 | 22.2 | 233 | 22.2 | 1,797 | 17.2 |
| 45–54 | 897 | 15.5 | 443 | 23.9 | 385 | 29.5 | 116 | 26.6 | 295 | 28.1 | 2,136 | 20.5 |
| 55–64 | 822 | 14.2 | 501 | 27.0 | 367 | 28.1 | 93 | 21.3 | 245 | 23.3 | 2,028 | 19.4 |
| 65–74 | 671 | 11.7 | 463 | 25.0 | 185 | 14.2 | 28 | 6.4 | 114 | 10.9 | 1,461 | 14.0 |
| **Sex** | | | | | | | | | | | | |
| Male | 2,142 | 37.0 | 1,355 | 73.1 | 974 | 74.7 | 327 | 74.8 | 867 | 82.7 | 5,665 | 54.3 |
| Female | 3,654 | 63.0 | 498 | 26.9 | 330 | 25.3 | 110 | 25.2 | 182 | 17.3 | 4,774 | 45.7 |
| **BMI (kg/m$^2$)** | | | | | | | | | | | | |
| ≤ 18.4 | 920 | 15.9 | 110 | 5.9 | 127 | 9.7 | 36 | 8.2 | 103 | 9.8 | 1,296 | 12.4 |
| 18.5–24.9 | 4,116 | 71.0 | 1,236 | 66.7 | 869 | 66.6 | 293 | 67.1 | 689 | 65.7 | 7,203 | 69.0 |
| 25.0–29.9 | 616 | 10.6 | 425 | 23.0 | 255 | 19.6 | 89 | 20.4 | 215 | 20.5 | 1,600 | 15.3 |
| ≥ 30.0 | 144 | 2.5 | 82 | 4.4 | 53 | 4.1 | 19 | 4.3 | 42 | 4.0 | 340 | 3.3 |
| **Marital status** | | | | | | | | | | | | |
| Single | 2,851 | 49.2 | 370 | 20.0 | 425 | 32.6 | 106 | 24.2 | 290 | 27.6 | 4,042 | 38.7 |
| Married | 2,629 | 45.3 | 1,333 | 71.9 | 746 | 57.2 | 291 | 66.6 | 672 | 64.1 | 5,671 | 54.3 |
| Widowed/divorced | 316 | 5.5 | 150 | 8.1 | 133 | 10.2 | 40 | 9.2 | 87 | 8.3 | 726 | 7.0 |
| **30-day alcohol use** | | | | | | | | | | | | |
| Present | 2,598 | 44.8 | 1,277 | 68.9 | 899 | 68.9 | 294 | 67.3 | 799 | 76.2 | 5,867 | 56.2 |
| **Income (USD /year)** | | | | | | | | | | | | |
| ≤ 9,999 | 286 | 4.9 | 45 | 2.4 | 38 | 2.9 | 13 | 3.0 | 23 | 2.2 | 405 | 3.9 |
| 10,000–39,999 | 1,286 | 22.2 | 452 | 24.4 | 322 | 24.7 | 78 | 17.8 | 185 | 17.7 | 2,323 | 22.3 |
| 40,000–79,999 | 1,678 | 29.0 | 632 | 34.1 | 443 | 34.0 | 143 | 32.7 | 386 | 36.8 | 3,282 | 31.4 |
| 80,000–119,999 | 753 | 13.0 | 282 | 15.2 | 179 | 13.7 | 85 | 19.5 | 215 | 20.5 | 1,514 | 14.5 |
| ≥ 120,000 | 321 | 5.5 | 138 | 7.5 | 89 | 6.8 | 44 | 10.1 | 120 | 11.4 | 712 | 6.8 |
| Unknown | 1,472 | 25.4 | 304 | 16.4 | 233 | 17.9 | 74 | 16.9 | 120 | 11.4 | 2,203 | 21.1 |
| **Educational level** | | | | | | | | | | | | |
| Less than high school | 197 | 3.4 | 52 | 2.8 | 56 | 4.3 | 13 | 3.0 | 29 | 2.8 | 347 | 3.3 |
| High school graduate | 1,689 | 29.2 | 528 | 28.5 | 414 | 31.7 | 132 | 30.2 | 272 | 25.9 | 3,035 | 29.1 |
| College or associate's degree | 1,185 | 20.4 | 354 | 19.1 | 272 | 20.9 | 83 | 19.0 | 144 | 13.7 | 2,038 | 19.5 |
| Bachelor's degree | 2,396 | 41.3 | 831 | 44.9 | 512 | 39.3 | 186 | 42.5 | 553 | 52.7 | 4,478 | 42.9 |
| Master's or doctoral degree | 329 | 5.7 | 88 | 4.7 | 50 | 3.8 | 23 | 5.3 | 51 | 4.9 | 541 | 5.2 |
| **Routine dental checkup** | | | | | | | | | | | | |
| Present | 2,864 | 49.4 | 982 | 53.0 | 564 | 43.3 | 230 | 52.6 | 572 | 54.5 | 5,212 | 49.9 |
| **Use of other tobacco products** | | | | | | | | | | | | |
| Present | | | | | 103 | 7.9 | 52 | 11.9 | 219 | 20.9 | 374 | 3.6 |
| **Secondhand exposure** | | | | | | | | | | | | |
| To combustible cigarettes | 3,034 | 52.3 | 1123 | 60.6 | 999 | 76.6 | 293 | 67.0 | 850 | 81.0 | 6,299 | 60.3 |
| To heated tobacco products or e-cigarettes | 1,573 | 27.1 | 628 | 33.9 | 503 | 38.6 | 274 | 62.7 | 668 | 63.7 | 3,646 | 34.9 |
| **Smoking pack-years** | | | | | | | | | | | | |

(*Continued*)

**Table 1.** (Continued)

| | Never user | | Former user | | Current user | | | | | | Total | |
|---|---|---|---|---|---|---|---|---|---|---|---|---|
| | | | | | N = 2,790 | | | | | | | |
| | | | | | Only combustible cigarettes | | Only heated tobacco products | | Combined use | | | |
| | N = 5,796 | | N = 1,853 | | N = 1,304 | | N = 437 | | N = 1,049 | | N = 10,439 | |
| | n | % | n | % | n | % | n | % | n | % | n | % |
| ≤ 5 | | | | | 381 | 29.2 | 179 | 41.0 | 238 | 22.7 | 798 | 7.6 |
| 6–10 | | | | | 146 | 11.2 | 55 | 12.6 | 108 | 10.3 | 309 | 3.0 |
| 11–20 | | | | | 284 | 21.8 | 72 | 16.5 | 191 | 18.2 | 547 | 5.2 |
| 21–30 | | | | | 214 | 16.4 | 55 | 12.6 | 188 | 17.9 | 457 | 4.4 |
| 31–40 | | | | | 126 | 9.7 | 43 | 9.8 | 133 | 12.7 | 302 | 2.9 |
| 41–50 | | | | | 72 | 5.5 | 19 | 4.3 | 84 | 8.0 | 175 | 1.7 |
| ≥ 51 | | | | | 81 | 6.2 | 14 | 3.2 | 107 | 10.2 | 202 | 1.9 |
| **Comorbidities** | | | | | | | | | | | | |
| History of hypertension | 588 | 10.1 | 495 | 26.7 | 231 | 17.7 | 89 | 20.4 | 221 | 21.1 | 1,624 | 15.6 |
| History of diabetes mellitus | 150 | 2.6 | 189 | 10.2 | 106 | 8.1 | 37 | 8.5 | 99 | 9.4 | 581 | 5.6 |
| History of bronchitis or pneumonia | 91 | 1.6 | 34 | 1.8 | 21 | 1.6 | 16 | 3.7 | 36 | 3.4 | 198 | 1.9 |
| History of heart diseases | 39 | 0.7 | 54 | 2.9 | 21 | 1.6 | 7 | 1.6 | 40 | 3.8 | 161 | 1.5 |
| History of stroke | 26 | 0.4 | 32 | 1.7 | 9 | 0.7 | 8 | 1.8 | 16 | 1.5 | 91 | 0.9 |
| **Outcomes** | | | | | | | | | | | | |
| **Prevalence of periodontal disease** | | | | | | | | | | | | |
| Overall | 463 | 8.0 | 346 | 18.7 | 200 | 15.3 | 66 | 15.1 | 204 | 19.4 | 1,279 | 12.3 |
| Under treatment | 192 | 3.3 | 172 | 9.3 | 92 | 7.1 | 32 | 7.3 | 88 | 8.4 | 576 | 5.5 |

respectively). In the second sensitivity analysis adjusting for DAG-based confounder, similar association was observed (S2 Table). In both main and sensitivity analyses, all VIFs were less than 2.5; and no problematic multicollinearity was observed (S3 Table).

## Discussion

Our results suggested that HTP use was significantly associated with the prevalence of self-reported periodontal disease after adjustment of 12 confounders, compared with never-users. Furthermore, former users, combustible tobacco product users, and combined users showed the same association as HTP users. The results were consistent even when the outcome was "active" in the first sensitivity analysis and the DAG-based covariate models were constructed in the second sensitivity analysis.

The growing number of people using HTPs is a critical global public health issue, as is the use of combustible tobacco. The prevalence of HTP use and smoking patterns in Japan were influenced by the early introduction and adoption of HTPs in this country [4]. In fact, according to the 2018 Japanese National Health and Nutrition Survey, male and female HTP users accounted for 30.6% (HTP only 22.1%; combined use of combustible tobacco products and HTPs 8.5%) and 23.6% (HTP only 14.8%; combined use 8.8%) of habitual tobacco product users, respectively [23]. Similar to the Japanese trends, the reported use of HTPs in the United States was up to 2.4% in 2018, and appeared to be increasing [24]. The fact that a growing number of people are using HTPs is a relevant public health problem both in Japan and the United States. On the other hand, concerning periodontal disease—also an important public health issue—evidence of HTPs is scarce, although there have been numerous studies that demonstrate

**Table 2. Results of multivariable modified Poisson regression analysis to estimate the prevalence ratios for overall periodontal disease and periodontal disease under treatment.**

| | Overall periodontal disease | | | | Periodontal disease under treatment | | | |
|---|---|---|---|---|---|---|---|---|
| | PR | | 95%CI | | PR | | 95%CI | |
| **Smoking status** | | | | | | | | |
| **Never user** | 1.00 | | (Reference) | | 1.00 | | (Reference) | |
| **Former user** | 1.56 | 1.35 | – | 1.80 | 1.85 | 1.48 | – | 2.30 |
| **Current user** | | | | | | | | |
| Combustible cigarette use | 1.29 | 1.03 | – | 1.62 | 1.77 | 1.27 | – | 2.46 |
| HTP use | 1.43 | 1.08 | – | 1.88 | 1.82 | 1.21 | – | 2.74 |
| Combined use | 1.55 | 1.20 | – | 1.99 | 1.82 | 1.26 | – | 2.63 |
| **Demographics** | | | | | | | | |
| **Age** | | | | | | | | |
| 18–24 | 0.63 | 0.47 | – | 0.84 | 0.63 | 0.39 | – | 1.03 |
| 25–34 | 0.62 | 0.46 | – | 0.84 | 0.65 | 0.41 | – | 1.05 |
| 35–44 | 1.00 | | (Reference) | | 1.00 | | (Reference) | |
| 45–54 | 1.51 | 1.25 | – | 1.82 | 1.30 | 0.96 | – | 1.76 |
| 55–64 | 2.12 | 1.77 | – | 2.55 | 2.16 | 1.64 | – | 2.84 |
| 65–74 | 1.81 | 1.47 | – | 2.22 | 1.73 | 1.28 | – | 2.33 |
| **Sex** | | | | | | | | |
| Male | 1.00 | | (Reference) | | 1.00 | | (Reference) | |
| Female | 1.17 | 1.03 | – | 1.33 | 1.21 | 0.99 | – | 1.47 |
| **BMI** | | | | | | | | |
| ≤ 18.4 | 1.02 | 0.85 | – | 1.22 | 0.75 | 0.54 | – | 1.03 |
| 18.5–24.9 | | | (Reference) | | | | (Reference) | |
| 25.0–29.9 | 0.98 | 0.86 | – | 1.12 | 1.10 | 0.90 | – | 1.34 |
| ≥ 30.0 | 1.21 | 0.96 | – | 1.52 | 1.61 | 1.16 | – | 2.25 |
| **Marital status** | | | | | | | | |
| Single | | | (Reference) | | | | (Reference) | |
| Married | 1.14 | 0.98 | | 1.32 | 1.40 | 1.10 | | 1.79 |
| Widowed/divorced | 1.19 | 0.97 | | 1.45 | 1.50 | 1.09 | | 2.08 |
| **30-day alcohol use** | | | | | | | | |
| Absent | 1.00 | | (Reference) | | 1.00 | | (Reference) | |
| Present | 1.07 | 0.95 | – | 1.19 | 0.94 | 0.79 | – | 1.11 |
| **Income (USD /year)** | | | | | | | | |
| ≤ 9,999 | 1.26 | 0.95 | – | 1.66 | 1.59 | 1.06 | – | 2.38 |
| 10,000–39,999 | 1.12 | 0.98 | – | 1.28 | 1.15 | 0.94 | – | 1.40 |
| 40,000–79,999 | 1.00 | | (Reference) | | 1.00 | | (Reference) | |
| 80,000–119,999 | 0.81 | 0.68 | – | 0.95 | 0.67 | 0.52 | – | 0.88 |
| ≥ 120,000 | 0.78 | 0.63 | – | 0.97 | 0.81 | 0.59 | – | 1.09 |
| Secret | 0.80 | 0.68 | – | 0.94 | 0.71 | 0.55 | – | 0.92 |
| **Educational level** | | | | | | | | |
| Less than high school | 1.33 | 1.02 | – | 1.73 | 0.93 | 0.56 | – | 1.55 |
| High school graduate | | | (Reference) | | | | (Reference) | |
| College or associate's degree | 0.80 | 0.69 | – | 0.93 | 0.87 | 0.70 | – | 1.09 |
| Bachelor's degree | 0.86 | 0.76 | – | 0.97 | 0.84 | 0.69 | – | 1.02 |
| Master's or doctoral degree | 0.80 | 0.61 | – | 1.05 | 0.83 | 0.55 | – | 1.25 |
| **Routine dental checkup** | | | | | | | | |
| Absent | | | (Reference) | | | | (Reference) | |

*(Continued)*

**Table 2.** (Continued)

| | | Overall periodontal disease | | | Periodontal disease under treatment | | |
|---|---|---|---|---|---|---|---|
| | | PR | 95%CI | | PR | 95%CI | |
| | Present | 1.51 | 1.35 – | 1.68 | 6.22 | 4.85 – | 7.98 |
| **Use of other tobacco products** | | | | | | | |
| | Absent | 1.00 | (Reference) | | 1.00 | (Reference) | |
| | Present | 1.29 | 1.05 – | 1.60 | 1.13 | 0.81 – | 1.59 |
| **Secondhand exposure to combustible cigarettes from others** | | | | | | | |
| | Absent | 1.00 | (Reference) | | 1.00 | (Reference) | |
| | Present | 1.24 | 1.10 – | 1.40 | 1.08 | 0.90 – | 1.29 |
| **Secondhand exposure to heated tobacco products or e-cigarettes from others** | | | | | | | |
| | Absent | 1.00 | (Reference) | | 1.00 | (Reference) | |
| | Present | 1.07 | 0.95 – | 1.20 | 1.04 | 0.87 – | 1.25 |
| **Comorbidities** | | | | | | | |
| **History of hypertension** | | | | | | | |
| | Absent | 1.00 | (Reference) | | 1.00 | (Reference) | |
| | Present | 1.25 | 1.11 – | 1.41 | 1.19 | 1.00 – | 1.43 |
| **History of diabetes mellitus** | | | | | | | |
| | Absent | 1.00 | (Reference) | | 1.00 | (Reference) | |
| | Present | 1.37 | 1.18 – | 1.60 | 1.59 | 1.28 – | 1.99 |
| **History of bronchitis or pneumonia** | | | | | | | |
| | Absent | 1.00 | (Reference) | | 1.00 | (Reference) | |
| | Present | 1.85 | 1.45 – | 2.35 | 2.08 | 1.47 – | 2.95 |
| **History of heart diseases** | | | | | | | |
| | Absent | 1.00 | (Reference) | | 1.00 | (Reference) | |
| | Present | 1.27 | 0.98 – | 1.63 | 1.15 | 0.78 – | 1.70 |
| **History of stroke** | | | | | | | |
| | Absent | 1.00 | (Reference) | | 1.00 | (Reference) | |
| | Present | 1.16 | 0.83 – | 1.63 | 1.22 | 0.72 – | 2.08 |
| **Smoking pack-years** | | | | | | | |
| | $\leq 5$ | 1.00 | (Reference) | | 1.00 | (Reference) | |
| | 6–10 | 1.10 | 0.80 – | 1.52 | 1.26 | 0.81 – | 1.98 |
| | 11–20 | 1.05 | 0.80 – | 1.37 | 0.79 | 0.51 – | 1.22 |
| | 21–30 | 0.97 | 0.74 – | 1.28 | 0.93 | 0.61 – | 1.43 |
| | 31–40 | 1.10 | 0.83 – | 1.46 | 0.96 | 0.63 – | 1.47 |
| | 41–50 | 1.31 | 0.97 – | 1.77 | 1.61 | 1.07 – | 2.42 |
| | $\geq 51$ | 1.14 | 0.83 – | 1.56 | 1.09 | 0.66 – | 1.77 |

Note. PR, prevalence ratio; CI, confidence interval; HTP, heated tobacco products; e-cig, electronic cigarette; BMI, body mass index.

the negative impacts of smoking and the association with periodontal disease [10]. Given these facts, this study clarified the association of HTP use, as well as combustible cigarette use and combined use, with periodontal disease using the "fertile" market of HTPs: the Japanese population [4]. To our best knowledge, this is the first study elucidating this association.

Some potential mechanisms might be considered to explain our results. The first is regarding the cytotoxic effect of HTPs on periodontal tissues. For example, emissions of carbonyl aldehydes from HTPs activate the autoimmune system, which causes a breakdown of periodontal matrix and bone loss [25]. In addition, aerosols containing nicotine accelerate oral myoblast differentiation, resulting in a worsening periodontal environment due to delayed

wound healing [25]. Along with those known factors, other potential chemical substances that occur at higher levels in HTPs than combustible tobacco products [3] might contribute to the prevalence of periodontal disease. Our results showed that PRs of HTPs for periodontal disease were comparable to that of combustible cigarettes (1.29 [1.03–1.62] vs. 1.43 [1.08–1.88] for overall periodontal disease; 1.77 [1.27–2.46] vs. 1.82 [1.26–2.63] for periodontal disease under treatment). The second explanation concerns the detrimental effect of the product use on the oral microbiome. Nicotine, as well as smoke from combustible cigarettes, causes the development of dysbiosis in the mouth and periodontal inflammation [26]. Although HTPs contain lower levels of nicotine and some known harmful toxicants than combustible cigarettes [3], habitual use of HTPs might contribute to the destruction of oral microbiomes.

A strength of our study is that we adjusted respondents' biological features (age, sex, and BMI), socioeconomic status (marital status, alcohol use, income, and educational level), comorbidities (hypertension, diabetes, acute respiratory infections, heart diseases, and stroke), routine dental checkup, and smoking-related conditions (other tobacco product use, second-hand exposure to combustible cigarettes or new tobacco products, and smoking pack-years) in the logistic regression model. Moreover, our study had a large sample size (10,439 included respondents and 1,486 HTP users) and a sufficient number of outcomes (1,279 for overall periodontal disease; 576 for active periodontal diseases) to produce reliable estimations in the multivariable logistic regression models [27]. Despite these strengths, our study had several limitations. First, this survey was based on a self-reported questionnaire. Therefore, there might be an underestimation of exposures [28]. Also, the outcome was defined as visits to the dentist for the treatment of periodontal disease; therefore, localization, severity, or subtypes of the disease could not be discerned. Third, this was an observational study; therefore, residual or unmeasured confounders may exist. Fourth, this study used a cross-sectional design, and could not refer to longitudinal association or causation. Fifth, this survey does not represent the general public in Japan; it is still unclear that our results are fully applicable to the Japanese. Further studies that compensate for the above-mentioned limitations are warranted.

The market for HTPs and the number of products continues to grow. HTPs are now available in more than 30 countries [29], and their use may continue to increase globally. Our results might provide meaningful information for users, healthcare professionals, and policymakers about a potential association of HTPs to oral health.

In conclusion, our study indicated that HTP use, as well as combustible cigarette and combined product use, were associated with a higher prevalence of periodontal diseases compared to non-users after adjusting for 12 confounders. Our results suggest that HTPs might be harmful to oral health, similar to the effects of combustible cigarettes. By exposing the association of HTP product use with periodontal disease, this study may lead epidemiology and public health researchers to examine other implications of the use of these new products. Also, further research is needed to confirm our results and investigate the longitudinal associations between HTP use and periodontal disease.

## Supporting information

**S1 Table. Definition of exposure.**
(DOCX)

**S2 Table. Results of sensitivity analysis: Multivariable modified Poisson regression analysis to estimate the prevalence ratios for overall periodontal disease and periodontal disease under treatment after adjusting for DAG-based confounders.**
(DOCX)

**S3 Table. Variance inflation factors to evaluate multicollinearity of each variable for main and sensitivity analyses.**
(DOCX)

**S1 Fig. A directed acyclic graph of this study.**
(DOCX)

## Acknowledgments

We would like to thank Editage (http://www.editage.com) for language editing.

## Author Contributions

**Conceptualization:** Takashi Yoshioka, Takahiro Tabuchi.

**Data curation:** Takahiro Tabuchi.

**Formal analysis:** Takashi Yoshioka.

**Funding acquisition:** Takahiro Tabuchi.

**Investigation:** Takahiro Tabuchi.

**Methodology:** Takashi Yoshioka.

**Project administration:** Takahiro Tabuchi.

**Resources:** Takahiro Tabuchi.

**Software:** Takashi Yoshioka.

**Supervision:** Takahiro Tabuchi.

**Validation:** Takahiro Tabuchi.

**Visualization:** Takashi Yoshioka.

**Writing – original draft:** Takashi Yoshioka.

**Writing – review & editing:** Takahiro Tabuchi.

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
