## [Decision Letter · Decision Letter 0]

13 Nov 2020

PONE-D-20-24726

Combustible cigarettes, heated tobacco products, combined product use, and periodontal disease: A cross-sectional JASTIS study

PLOS ONE

Dear Dr. Tabuchi,

Thank you for submitting your manuscript to PLOS ONE. After careful consideration, we feel that it has merit but does not fully meet PLOS ONE’s publication criteria as it currently stands. Therefore, we invite you to submit a revised version of the manuscript that addresses the points raised during the review process.

Despite asking many people, I was only able to secure a single review for your paper.  Fortunately, the one review I have is of high quality, so I have decided to invite major revision based on that single review.

We look forward to receiving your revised manuscript.

Kind regards,

Stanton A. Glantz

Academic Editor

PLOS ONE

Journal Requirements:

Reviewers' comments:

Reviewer's Responses to Questions

**Comments to the Author**

1. Is the manuscript technically sound, and do the data support the conclusions?

Reviewer #1: Partly

2. Has the statistical analysis been performed appropriately and rigorously? 

Reviewer #1: No

3. Have the authors made all data underlying the findings in their manuscript fully available?

Reviewer #1: No

4. Is the manuscript presented in an intelligible fashion and written in standard English?

Reviewer #1: Yes

5. Review Comments to the Author

Reviewer #1: This manuscript reports associations between heated tobacco product use and self-reported periodontal disease in a large Japanese cross-sectional sample. However, additional details are needed and the analytic methods should be modified for this cross-sectional survey.

1. Kassebaum et al. estimated global prevalence of severe periodontitis at 7.4% not “20-50%” as cited in the paper.

2. The authors should include additional relevant works such as Kassebaum et al. 2017 and Schwendicke et al. 2018 on the global burden of disease in the Introduction and/or Discussion sections.

3. What “(1)… [inconsistencies] with the information they had provided in the earlier surveys (2015 to 2018)” were used to exclude respondents? How many surveys were excluded for reasons (1) or (2)?

4. The authors refer to covariates as “12 confounders” but they have not shown them from the literature or empirically to be confounders. So it is clearer to refer to them as covariates or “potential” confounders.

5. Are there any backdoor correlations induced by overadjustment (ie, adjusting for a covariate that is a result of HTP and a cause of periodontal disease)? A directed acyclic graph (DAG) for hypothesized relationships would be helpful to better understand the hypothesized relationships. For example, is adjusting for exposure to other tobacco product use, secondhand HTP smoke, or diabetes problematic because HTP use relates to THEM which relates to periodontal disease? (in contrast, a confounder would relate TO HTP use and to periodontal disease, not result FROM HTP use.)

6. More details about the design of the cross-sectional study should be provided here rather than just citing Tabuchi et al 2019. Was this a random sample of an existing internet panel? What internet panel provider was used (Rakuten)? What was the response rate?

7. The paper (Tabuchi et al 2019) they cite stated “Respondents of an internet study are not representative of the general population, so we conducted statistical adjustment to account for bias” and “The response rate in the follow-up survey was also problematic, given that non-responders differ in a number of ways from the respondents in the survey.” Were survey weights (eg iterative proportional weights) used to attempt to reduce these biases as in ref#11 (Tabuchi et al 2019)? If so, this manuscript itself needs to provide enough basic info about the study design for readers of this paper to understand the design and the appropriateness of the analyses. If not, why were IPWs not used to try to reduce these biases?

8. Why did the authors use logistic regression models for odds ratios (which are appropriate in case-control studies) instead of the more appropriate log-binomial or Poisson regression models for prevalence ratios which can also be performed with Stata?

9. Add multicollinearity assessment for the models which have some covariates that may be correlated too much with each other.

10. The Results reports “prevalence for combustible tobacco product users, HTP users, and combined product”. This is “periodontal prevalence for combustible…” not prevalence of product use, right?

11. Why would former users have stronger relationship to periodontal disease than current use types? Why does having a routine dental visit relate to periodontal disease – could this be reverse causation? Why do you claim the chance of reverse causation is low in the Discussion?

12. The title of Table 2 should list the response variable (i.e. periodontal disease).

13. An editorial item – a patient can be ambulant or ambulatory; a treatment itself is not ambulant or ambulatory, but for ambulant or ambulatory patients.

References:

Kassebaum et al. JDR 2017

Schwendicke et al. J Clin Periodontol 2018

Tabuchi et al. J Epidemiol 2019

6. PLOS authors have the option to publish the peer review history of their article (what does this mean?). If published, this will include your full peer review and any attached files.

Reviewer #1: **Yes: **Stuart Gansky

---

## [Author Response · Author response to Decision Letter 0]

25 Jan 2021

Comments from Reviewer #1: 

This manuscript reports associations between heated tobacco product use and self-reported periodontal disease in a large Japanese cross-sectional sample. However, additional details are needed and the analytic methods should be modified for this cross-sectional survey.

Response: We thank Professor Stuart Gansky, for his insightful comments and grateful for the review. We have incorporated changes in the manuscript to reflect of the suggestions provided by the reviewer. The changes were recorded by yellow markers. We hope that the revised manuscript is now suitable for publication. 

Comment 1: 

Kassebaum et al. estimated global prevalence of severe periodontitis at 7.4% not “20-50%” as cited in the paper.

Response: We appreciate this observation and apologize for the incorrect information. We agree that 7.4% describes the global trends of severe periodontitis. Therefore, we added a new reference (#8), and changed the manuscript below.

Changes: 

Introduction

Lines 59–61

Periodontal disease, which is characterized by the loss of periodontal ligament tissue and the collapse of surrounding alveolar bone tissue, is prevalent (7.4%) globally [8].

References

Lines 326–333

8. Kassebaum NJ, Smith AGC, Bernabé E, Fleming TD, Reynolds AE, Vos T, Murray CJL, Marcenes W; GBD 2015 Oral Health Collaborators. Global, Regional, and National Prevalence, Incidence, and Disability-Adjusted Life Years for Oral Conditions for 195 Countries, 1990-2015: A Systematic Analysis for the Global Burden of Diseases, Injuries, and Risk Factors. J Dent Res. 2017;96(4):380-387. 

9. Nazir MA. Prevalence of periodontal disease, its association with systemic diseases and prevention. Int J Health Sci (Qassim). 2017;11(2): 72-80.

Comment 2: The authors should include additional relevant works such as Kassebaum et al. 2017 and Schwendicke et al. 2018 on the global burden of disease in the Introduction and/or Discussion sections.

Response: We appreciate the reviewer suggestions of these important studies. As the reviewer instructed, we documented the global burden of disease referring to the suggested two papers.

Changes:

Introduction

Lines 63–67

Periodontal disease, especially severe chronic periodontal disease (SCP), is the second leading cause of disability-adjusted life-years (DALYs) associated with oral conditions [8]. The estimated global burden of SCP accounted for 3.5 million disability-adjusted life-years (DALYs) (95% uncertainty intervals [UI]: 1.4 to 7.2) with an increasing trend from 1995 to 2015 [8].

Lines 73–75

The global smoking-attributed burden of periodontal disease was also high, which accounted for 251,160 DALYs (95% UI: 190,721 to 324,241) among 186 countries in 2015 [11].

References

Lines 328–333

8. Kassebaum NJ, Smith AGC, Bernabé E, Fleming TD, Reynolds AE, Vos T, Murray CJL, Marcenes W; GBD 2015 Oral Health Collaborators. Global, Regional, and National Prevalence, Incidence, and Disability-Adjusted Life Years for Oral Conditions for 195 Countries, 1990-2015: A Systematic Analysis for the Global Burden of Diseases, Injuries, and Risk Factors. J Dent Res. 2017;96(4):380-387. 

Lines 337–339

11. Schwendicke F, Dörfer CE, Meier T. Global smoking-attributable burden of periodontal disease in 186 countries in the year 2015. J Clin Periodontol. 2018 Jan;45(1):2-14.

Comment 3: What “(1)… [inconsistencies] with the information they had provided in the earlier surveys (2015 to 2018)” were used to exclude respondents? How many surveys were excluded for reasons (1) or (2)?

Response: 

We apologize to the reviewer for our inaccurate expressions in the manuscript. Those who met reason (1) were respondents who participated in JASTIS two times or more and whose basic information, such as sex, was inconsistent. The number of such respondents was two in 2015–2018. They are a priori excluded; therefore, we should have described the reason (1) as an inclusion criterion. On the other hand, those who met reason (2) were all the excluded participants in the Fig 1, that is the exclusion criteria. The number of excluded respondents due to (2) was 561. To make it clearer, we distinguished the inclusion and exclusion criteria, and changed the manuscript as follows:

Changes:

Methods

Inclusion and exclusion criteria

Lines 110–116

This study included data from all respondents of the 2019 survey, except those whose responses were inconsistent with the information they had provided in the earlier surveys (2015 to 2018). On the other hand, this study excluded those whose responses contained other discrepancies. For example, we excluded surveys if the respondents chose the same answer number for all questions in a set of questions, or those who reported an amount of tobacco product use but had indicated that they had never used, or were only former users of, tobacco products.

Comment 4: The authors refer to covariates as “12 confounders” but they have not shown them from the literature or empirically to be confounders. So it is clearer to refer to them as covariates or “potential” confounders.

Response: 

We thank the reviewer for the insightful comment. We totally agree with the reviewer. Following the comment, we clearly distinguished the known confounders and potential confounders, and cited new references, which account for the confounding factors.

Changes:

Methods

Covariates

Lines 148–157 

Based on the previous studies regarding the association between combustible cigarettes/e-cigarettes and periodontal disease, we selected age, sex, body mass index (BMI), socioeconomic status (marital status, educational levels, household income), alcohol use [12,15,16], routine dental checkups [12,15], secondhand exposure of combustible cigarettes from others, secondhand aerosols of heated tobacco products from others [12], smoking pack-years [17], and comorbidities (hypertension, diabetes mellitus, heart disease) [17,18,19] as confounders. In addition to such known confounders, we further selected additional potential confounders (stroke, other tobacco product use) for constructing multivariable models. In total, we selected 12 participant characteristics confounders as covariates. These variables were categorized as follows: age (18–24 years, 25–34, 35–44, 45–54, 55–64, and 65–74), sex (male and female), …

Discussion

Lines 223–224 

Our results suggested that HTP use was significantly associated with the prevalence of self-reported periodontal disease after adjustment of potential 12 confounders, …

Lines 287–289 

In conclusion, our study indicated that HTP use, as well as combustible cigarette and combined product use, were associated with a higher prevalence of periodontal diseases compared to non-users after adjusting for 12 potential confounders.

References

Lines 352–369

15. Mundt T, Schwahn C, Mack F, Plzer I, Samietz S, Kocher T, et al. Risk indicators for missing teeth in working-age pomeranians - An evaluation of high-risk populations. J Public Health Dent. 2007;67(4):243-249. doi:10.1111/j.1752-7325.2007.00041.x

16. Hanioka T, Ojima M, Tanaka K, Aoyama H. Relationship between smoking status and tooth loss: Findings from national databases in Japan. J Epidemiol. 2007;17(4):125-132. doi:10.2188/jea.17.125

17. Dietrich T, Stosch U, Dietrich D, Kaiser W, Bernimoulin J-P, Joshipura K. Prediction of Periodontal Disease From Multiple Self-Reported Items in a German Practice-Based Sample. J Periodontol. 2007;78(7s):1421-1428. doi:10.1902/jop.2007.060212

18. Similä T, Auvinen J, Timonen M, Virtanen JI. Long-term effects of smoking on tooth loss after cessation among middle-aged Finnish adults: The Northern Finland Birth Cohort 1966 Study. BMC Public Health. 2016;16(1):1-8. doi:10.1186/s12889-016-3556-

19. Dietrich T, Walter C, Oluwagbemigun K, Bergmann M, Pischon T, Pischon N, et al. Smoking, smoking cessation, and risk of tooth loss: The EPIC-Potsdam study. J Dent Res. 2015;94(10):1369-1375. doi:10.1177/0022034515598961

Comment 5: Are there any backdoor correlations induced by overadjustment (ie, adjusting for a covariate that is a result of HTP and a cause of periodontal disease)? A directed acyclic graph (DAG) for hypothesized relationships would be helpful to better understand the hypothesized relationships. For example, is adjusting for exposure to other tobacco product use, secondhand HTP smoke, or diabetes problematic because HTP use relates to THEM which relates to periodontal disease? (in contrast, a confounder would relate TO HTP use and to periodontal disease, not result FROM HTP use.)

Response: 

We thank the reviewer for the insightful comment. Although our selection of confounders in the multivariable models was based on the previous studies, it is important to avoid biased estimation due to unnecessary variable adjustment. Following the reviewer’s instruction, we described a DAG (S1 Fig).

In our study, secondhand smoking and secondhand aerosols were measured as passive secondhand exposure from others. Therefore, we consider them as confounders, not intermediate factors. 

As described in the DAG, diabetes, stroke, and other tobacco use may be intermediate factors. Furthermore, respiratory infection, hypertension, and heart disease may be colliders. To estimate the possible effect of overadjustment bias and collider stratification bias on the main analysis, we performed an additional sensitivity analysis where confounders were selected based on the DAG and checked the consistency of the results in the main analysis. The results were almost the same as the main analysis, as described below.

Given these discussions, we changed the following sentences regarding secondhand smoking, secondhand aerosols, description of DAG, and an additional sensitivity analysis in the manuscript, and added a supplementary Figure and a Table (S1 Fig, S2 Table).

Changes:

Methods

Covariates

Lines 151–153

…secondhand exposure of combustible cigarettes from others, secondhand aerosols of heated tobacco products from others [12],

Lines 163–165 

…secondhand smoke exposure from others (exposure to smoke/aerosols from combustible cigarettes and/or e-cigarettes or HTPs by others),

Statistical analysis

Lines 173–183

Additionally, we performed two sensitivity analyses. In the first sensitivity analysis, we changed the definition of exposure from overall periodontal diseases to those under treatment. Furthermore, to validate the confounding selection in the main analysis, we performed a second sensitivity analysis which describes a directed acyclic graph (DAG), and constructed multivariable models based on confounders from the DAG (i.e. age, sex, BMI, educational level, routine dental checkup, secondhand exposure to combustible cigarettes from others, secondhand exposure to heated tobacco products or e-cigarettes from others, and smoking pack-years), and confirmed consistencies of the results. To consider the multicollinearity of variables in the multivariable models, we calculated variance inflation factors (VIF) in both the main and sensitivity analyses.

Discussion

Lines 226–228 

The results were consistent even when the outcome was “active” in the first sensitivity analysis and the DAG-based covariate models in the second sensitivity analysis.

Supporting information

S2 Table. Results of sensitivity analysis: multivariable modified Poisson regression analysis to estimate the prevalence ratios for overall periodontal disease and periodontal disease under treatment after adjusting for DAG-based confounders

　 　 Overall periodontal disease 　 Periodontal disease under treatment

　 PR 95%CI PR 95%CI

Smoking status 

Never user 1.00 (Reference) 1.00 (Reference)

Former user 1.62 1.40 － 1.86 1.91 1.54 － 2.38 

Current user 

Combustible cigarette use 1.41 1.13 － 1.77 1.95 1.40 － 2.71 

HTP use 1.64 1.24 － 2.16 2.14 1.42 － 3.23 

Combined use 1.84 1.45 － 2.34 2.18 1.53 － 3.11 

Demographics 

Age 

18–24 0.63 0.47 － 0.84 0.64 0.39 － 1.05 

25–34 0.62 0.45 － 0.84 0.64 0.40 － 1.04 

35–44 1.00 (Reference) 1.00 (Reference)

45–54 1.52 1.26 － 1.84 1.31 0.97 － 1.77 

55–64 2.27 1.90 － 2.72 2.30 1.75 － 3.03 

65–74 2.01 1.65 － 2.45 1.91 1.42 － 2.56 

Sex 

Male 1.00 (Reference) 1.00 (Reference)

Female 1.13 1.00 － 1.29 1.16 0.95 － 1.40 

BMI 

≤ 18.4 0.99 0.83 － 1.19 0.73 0.53 － 1.01 

18.5–24.9 (Reference) (Reference)

25.0–29.9 1.05 0.92 － 1.19 1.20 0.99 － 1.45 

≥ 30.0 1.43 1.13 － 1.81 1.95 1.40 － 2.72 

Marital status 

Single (Reference) (Reference)

Married 1.15 0.98 － 1.33 1.42 1.11 － 1.82 

Widowed/divorced 1.17 0.96 － 1.44 1.48 1.06 － 2.05 

30-day alcohol use 

Absent 1.00 (Reference) 1.00 (Reference)

Present 1.06 0.95 － 1.18 0.92 0.78 － 1.08 

Income (USD /year) 

≤ 9,999 1.32 1.00 － 1.74 1.71 1.14 － 2.57 

10,000–39,999 1.12 0.98 － 1.27 1.13 0.93 － 1.38 

40,000–79,999 1.00 (Reference) 1.00 (Reference)

80,000–119,999 0.78 0.67 － 0.91 0.67 0.52 － 0.87 

≥ 120,000 0.85 0.75 － 0.96 0.83 0.61 － 1.12 

Unknown 0.78 0.59 － 1.02 0.69 0.53 － 0.90 

Educational level 

Less than high school 1.39 1.07 － 1.81 1.04 0.63 － 1.71 

High school graduate (Reference) (Reference)

College or associate’s degree 0.78 0.67 － 0.91 0.85 0.68 － 1.06 

Bachelor's degree 0.85 0.75 － 0.96 0.83 0.68 － 1.00 

Master's or doctoral degree 0.78 0.59 － 1.02 0.79 0.53 － 1.19 

Routine dental checkup 

Absent (Reference) (Reference)

Present 1.53 1.38 － 1.71 6.34 4.95 － 8.13 

Secondhand exposure to combustible cigarettes from others 

Absent 1.00 (Reference) 1.00 (Reference)

Present 1.24 1.09 － 1.40 1.08 0.90 － 1.30 

Secondhand exposure to combustible cigarettes from others 

Absent 1.00 (Reference) 1.00 (Reference)

Present 1.09 0.96 － 1.22 1.05 0.88 － 1.27 

Smoking pack-years 

≤ 5 1.00 (Reference) 1.00 (Reference)

6–10 1.03 0.75 － 1.41 1.12 0.72 － 1.75 

11–20 0.94 0.73 － 1.23 0.68 0.44 － 1.05 

21–30 0.89 0.68 － 1.17 0.84 0.55 － 1.27 

31–40 1.02 0.77 － 1.36 0.88 0.57 － 1.34 

41–50 1.22 0.91 － 1.65 1.46 0.97 － 2.18 

≥ 51 1.09 0.80 － 1.49 　 1.01 0.62 － 1.63 

Note. DAG, directed acyclic graph; PR, prevalence ratio; CI, confidence interval; HTP, heated tobacco products; e-cigarette, electronic cigarette; BMI, body mass index

Comment 6: More details about the design of the cross-sectional study should be provided here rather than just citing Tabuchi et al 2019. Was this a random sample of an existing internet panel? What internet panel provider was used (Rakuten)? What was the response rate?

Response: 

We thank the reviewer for this critical observation. We have provided the detailed information about the sampling method, the survey panels, and the response rate.

Changes:

Methods

Study design and setting

Lines 95–107

Participants of the JASTIS were recruited via a survey panel provided from a major internet research agency in Japan (Rakuten Insight). The agency keeps about 2.3 million panelists and their socioeconomic status, such as education levels, household income, and marital status. The survey panel comprised those who were recruited initially via services of the Rakuten agency group. For the 2015–2018 baseline survey, participants were randomly sampled from the total panelists from the Rakuten Insight database. The follow-up surveys in other years were conducted from all respondents who previously participated in JASTIS. In the 2019 survey, 9,262 participants were recruited from the follow-up survey. The response rate was 62.5% (9,262 /14,825). In addition to the follow-up survey, the 2019 survey collected new participants (1,738) aged 15–24 from the panel because of the small population compared with other age groups. The additional survey was closed when the target number of respondents who had answered the questionnaire was met. In total, 11,000 respondents participated in JASTIS 2019. 

Comment 7: The paper (Tabuchi et al 2019) they cite stated “Respondents of an internet study are not representative of the general population, so we conducted statistical adjustment to account for bias” and “The response rate in the follow-up survey was also problematic, given that non-responders differ in a number of ways from the respondents in the survey.” Were survey weights (eg iterative proportional weights) used to attempt to reduce these biases as in ref#11 (Tabuchi et al 2019)? If so, this manuscript itself needs to provide enough basic info about the study design for readers of this paper to understand the design and the appropriateness of the analyses. If not, why were IPWs not used to try to reduce these biases?

Response: 

This study was designed to evaluate the cross-sectional association between HTP use and periodontal disease among JASTIS 2019. In this study, we focused on evaluating the association, not description; representativeness is essential in this case. In addition, IPW is calculated using data from the 2016 Comprehensive Survey of Living Conditions, which is different in 2019. For these reasons, we did not use IPW adjustment. It is indeed true that the participants of JASTIS 2019 do not represent the general public in Japan, and the limitation of our study is this lack of generalizability. We added the following sentences in the Discussion section.

Changes:

Discussion

Lines 279–280

Fifth, this survey does not represent the general public in Japan; it is still unclear that our results are fully applicable to the Japanese.

Comment 8: Why did the authors use logistic regression models for odds ratios (which are appropriate in case-control studies) instead of the more appropriate log-binomial or Poisson regression models for prevalence ratios which can also be performed with Stata?

Response: 

In this study, we used logistic regression models and presented odds ratios for outcome measures because it is commonly used in the epidemiological and clinical research. However, as the reviewer pointed out, odds ratio for not rare outcomes may induce overestimation of the risk. Therefore, we conducted a modified Poisson regression analysis and estimated prevalence ratios for each covariate. The results were almost the same as the logistic regressions. Given this analysis and the reviewer’s kind instruction, we changed all the statistical analyses from using logistic regression models to modified Poisson regression models.

Changes: 

Abstract

Lines 27–29

We estimated adjusted prevalence ratios (PRs) and confidence intervals (CIs) using multivariable modified Poisson regression analysis after adjusting for 12 confounders.

Lines 32–36

Compared with never-users, HTP use was significantly associated with the prevalence of self-reported periodontal diseases (PR 1.43, 95% CI 1.03–1.62). Moreover, former users, combustible cigarette users, and combined product users also showed significant associations (PR 1.56, 95% CI 1.35–1.80; PR 1.29, 95% CI 1.03–1.62; and PR 1.55, 95% CI 1.20–1.99, respectively).

Methods

Statistical analysis

Lines 170–173 

Second, multivariable modified Poisson regression analysis was performed to estimate the prevalence ratio (PR) and confidence interval (CI) for the prevalence of self-reported periodontal disease as the primary analysis, adjusting for the 12 covariates [21,22].

Results

Lines 205–210

The results of multivariable modified Poisson regression analyses, both in the main and sensitivity analyses, were shown in Table 2. After adjustment for the 12 confounders, the PR of combustible products, HTPs, and combined use presented statistically significant associations (PR [95% CI]: 1.29 [1.03–1.62]; 1.43 [1.08–1.88]; and 1.55 [1.20–1.99]) compared with never-users, respectively. Moreover, former users were also significantly associated with the outcome (1.56 [1.35–1.80]). 

Lines 212–215

…similar significant associations were consistently found among former users, combustible tobacco product users, HTP users, and combined product users (PR [95% CI]: 1.85[1.48–2.30]; 1.77 [1.27–2.46]; 1.82 [1.21–2.74]; and 1.82 [1.26–2.63], respectively).

References 

Lines 374–377

21. Holmberg MJ, Andersen LW. Estimating Risk Ratios and Risk Differences Alternatives to Odds Ratios. 2021:2020-2021. doi:10.1093/aje/kwi188 

22. Zou G. A Modified Poisson Regression Approach to Prospective Studies with Binary Data. 2004;159(7):702-706. doi:10.1093/aje/kwh090

Table 2. Results of multivariable modified Poisson regression analysis to estimate the prevalence ratios for overall periodontal disease and periodontal disease under treatment

　 Overall periodontal disease 　 Periodontal disease under treatment

　 PR 95%CI 　 PR 95%CI

Smoking status 

Never user 1.00 (Reference) 1.00 (Reference)

Former user 1.56 1.35 － 1.80 1.85 1.48 － 2.30 

Current user 

Combustible cigarette use 1.29 1.03 － 1.62 1.77 1.27 － 2.46 

HTP use 1.43 1.08 － 1.88 1.82 1.21 － 2.74 

Combined use 1.55 1.20 － 1.99 1.82 1.26 － 2.63 

Demographics 

Age 

18–24 0.63 0.47 － 0.84 0.63 0.39 － 1.03 

25–34 0.62 0.46 － 0.84 0.65 0.41 － 1.05 

35–44 1.00 (Reference) 1.00 (Reference)

45–54 1.51 1.25 － 1.82 1.30 0.96 － 1.76 

55–64 2.12 1.77 － 2.55 2.16 1.64 － 2.84 

65–74 1.81 1.47 － 2.22 1.73 1.28 － 2.33 

Sex 

Male 1.00 (Reference) 1.00 (Reference)

Female 1.17 1.03 － 1.33 1.21 0.99 － 1.47 

BMI 

≤ 18.4 1.02 0.85 － 1.22 0.75 0.54 － 1.03 

18.5–24.9 (Reference) (Reference)

25.0–29.9 0.98 0.86 － 1.12 1.10 0.90 － 1.34 

≥ 30.0 1.21 0.96 － 1.52 1.61 1.16 － 2.25 

Marital status 

Single (Reference) (Reference)

Married 1.14 0.98 － 1.32 1.40 1.10 － 1.79 

Widowed/divorced 1.19 0.97 － 1.45 1.50 1.09 － 2.08 

30-day alcohol use 

Absent 1.00 (Reference) 1.00 (Reference)

Present 1.07 0.95 － 1.19 0.94 0.79 － 1.11 

Income (USD /year) 

≤ 9,999 1.26 0.95 － 1.66 1.59 1.06 － 2.38 

10,000–39,999 1.12 0.98 － 1.28 1.15 0.94 － 1.40 

40,000–79,999 1.00 (Reference) 1.00 (Reference)

80,000–119,999 0.81 0.68 － 0.95 0.67 0.52 － 0.88 

≥ 120,000 0.78 0.63 － 0.97 0.81 0.59 － 1.09 

Secret 0.80 0.68 － 0.94 0.71 0.55 － 0.92 

Educational level 

Less than high school 1.33 1.02 － 1.73 0.93 0.56 － 1.55 

High school graduate (Reference) (Reference)

College or associate’s degree 0.80 0.69 － 0.93 0.87 0.70 － 1.09 

Bachelor's degree 0.86 0.76 － 0.97 0.84 0.69 － 1.02 

Master's or doctoral degree 0.80 0.61 － 1.05 0.83 0.55 － 1.25 

Routine dental checkup 

Absent (Reference) (Reference)

Present 1.51 1.35 － 1.68 6.22 4.85 － 7.98 

Use of other tobacco products 

Absent 1.00 (Reference) 1.00 (Reference)

Present 1.29 1.05 － 1.60 1.13 0.81 － 1.59 

Secondhand exposure to combustible cigarettes from others 

Absent 1.00 (Reference) 1.00 (Reference)

Present 1.24 1.10 － 1.40 1.08 0.90 － 1.29 

Secondhand exposure to heated tobacco products or e-cigarettes from others 

Absent 1.00 (Reference) 1.00 (Reference)

Present 1.07 0.95 － 1.20 1.04 0.87 － 1.25 

Comorbidities 

History of hypertension 

Absent 1.00 (Reference) 1.00 (Reference)

Present 1.25 1.11 － 1.41 1.19 1.00 － 1.43 

History of diabetes mellitus 

Absent 1.00 (Reference) 1.00 (Reference)

Present 1.37 1.18 － 1.60 1.59 1.28 － 1.99 

History of bronchitis or pneumonia 

Absent 1.00 (Reference) 1.00 (Reference)

Present 1.85 1.45 － 2.35 2.08 1.47 － 2.95 

History of heart diseases 

Absent 1.00 (Reference) 1.00 (Reference)

Present 1.27 0.98 － 1.63 1.15 0.78 － 1.70 

History of stroke 

Absent 1.00 (Reference) 1.00 (Reference)

Present 1.16 0.83 － 1.63 1.22 0.72 － 2.08 

Smoking pack-years 

≤ 5 1.00 (Reference) 1.00 (Reference)

6–10 1.10 0.80 － 1.52 1.26 0.81 － 1.98 

11–20 1.05 0.80 － 1.37 0.79 0.51 － 1.22 

21–30 0.97 0.74 － 1.28 0.93 0.61 － 1.43 

31–40 1.10 0.83 － 1.46 0.96 0.63 － 1.47 

41–50 1.31 0.97 － 1.77 1.61 1.07 － 2.42 

≥ 51 1.14 0.83 － 1.56 1.09 0.66 － 1.77 

Note. PR, prevalence ratio; CI, confidence interval; HTP, heated tobacco products; e-cigarette, electronic cigarette; BMI, body mass index

Comment 9: Add multicollinearity assessment for the models which have some covariates that may be correlated too much with each other.

Response: 

Thank you for your feedback on this area. We calculated variance inflation factor (VIF), and constructed a Table in the supporting information (S3 Table). As a result, the mean VIF was 1.42, and no variables exceeded 2.5 in VIF. Similarly, we calculated VIF in the second sensitivity analysis, and no VIFs exceeded 2.5.

Changes:

Methods

Statistical analysis

Lines 181–183 

To consider the multicollinearity of variables in the multivariable models, we calculated variance inflation factors (VIF) in both the main and sensitivity analyses.

Supporting information

S3 Table. Variance inflation factors to evaluate multicollinearity of each variable for main and sensitivity analyses

Variable VIF

　 Main analysis 　 Sensitivity analysis

Smoking status 　 　

Former user 1.33 1.32

Current user 

Combustible cigarette use 2.26 2.2

HTP use 1.39 1.35

Combined use 2.43 2.23

Age 

18-24 2.21 2.21

25-34 1.56 1.56

45-54 1.81 1.8

55-64 1.97 1.9

65-74 1.94 1.82

Sex 

Female 1.43 1.42

BMI 

≤ 18.4 1.09 1.04

25.0-29.9 1.12 1.09

≥ 30.0 1.07 1.09

Marital status 

Married 1.87 1.87

Widowed/divorced 1.34 1.34

30-day alcohol use 

Present 1.19 1.19

Income (USD /year) 

≤ 9,999 1.15 1.15

10,000-39,999 1.44 1.43

80,000-119,999 1.28 1.28

 ≥ 120000 1.17 1.17

Unknown 1.43 1.43

Educational level 

Less than high school 1.1 1.09

College or associate’s degree 1.42 1.42

Bachelor's degree 1.52 1.52

Master's or doctoral degree 1.19 1.19

Routine dental checkup 

Present 1.07 1.07

Use of other tobacco products 

Present 1.18 

Secondhand exposure to combustible cigarettes from others 

Present 1.27 1.27

Secondhand exposure to HTPs or e-cigarettes from others 

Present 1.33 1.32

Smoking pack-years 

6–10 1.36 1.36

11–20 1.66 1.64

21–30 1.58 1.57

31–40 1.4 1.4

41–50 1.25 1.24

≥ 51 1.3 1.3

Comorbidities 

History of hypertension 

Present 1.26 

History of diabetes mellitus 

Present 1.13 

History of bronchitis or pneumonia 

Present 1.04 

History of heart diseases 

Present 1.09 

History of stroke 

Present 1.06 

Mean VIF 1.42 　 1.45

Note. VIF, variance inflation factors; HTP, heated tobacco products; BMI, body mass index; e-cigarettes, electronic cigarettes; 

Comment 10: The Results reports “prevalence for combustible tobacco product users, HTP users, and combined product”. This is “periodontal prevalence for combustible…” not prevalence of product use, right? 

Response: 

We thank the reviewer for making this observation. The reviewer’s correction is exactly what we would like to present. Therefore, we inserted the term “periodontal” in the following sentence on the Result section.

Changes:

Results

Lines 195–198 

The overall prevalence of periodontal disease was 12.3% (1,279/10,439); periodontal prevalence for combustible tobacco product users, HTP users, and combined product users was 15.3% (200/1,304), 15.1% (66/437), and 19.4% (204/1,049), respectively.

Comment 11: Why would former users have stronger relationship to periodontal disease than current use types? Why does having a routine dental visit relate to periodontal disease – could this be reverse causation? Why do you claim the chance of reverse causation is low in the Discussion?

Response: 

First, we would like to discuss why former users have a stronger relationship to periodontal disease than current users. We did not obtain the duration and the amount of tobacco product used among former uses, contrary to the current user. Therefore, we adjusted for smoking pack-years for current users only. For this reason, we considered that the point estimates among former users were higher than current users. 

Second, we discussed the relationship between routine dental checkups and periodontal disease. As the reviewer pointed out, those who had periodontal disease may consult dentists for routine dental checkup. On the contrary, people who undergo dental checkups are more likely to be diagnosed with periodontal disease by dentists. We considered that this association is bidirectional. 

Given the above discussions, we agreed with the reviewer’s indications and changed our consideration that the chance of reverse causation is low because of the presence of the above-mentioned bidirectional associations. Therefore, we changed the manuscript as follows.

Changes:

Discussion

Lines 277–279

Fourth, this study used a cross-sectional design, and we could not refer to longitudinal association or causation, although the possibility of reverse causation was low.

Comment 12: The title of Table 2 should list the response variable (i.e. periodontal disease).

Response: 

We agreed with the reviewer and rephrased the title of Table 2, as shown below.

Changes:

Table 2 Results of multivariable modified Poisson regression analysis to estimate the prevalence ratios for overall periodontal disease and periodontal disease under treatment.

Comment 13: An editorial item – a patient can be ambulant or ambulatory; a treatment itself is not ambulant or ambulatory, but for ambulant or ambulatory patients.

Response: 

We appreciate the reviewer’s insightful comment. To avoid the misuse of the terms, we deleted “ambulant” throughout the manuscript, and expressed “periodontal disease under treatment.”

Changes:

Methods

Main outcome measures

Lines 143–145

…and the latter one response was defined as “currently affected by periodontal disease under treatment (periodontal disease under treatment).”

Statistical analysis

Lines 174–175

…we changed the definition of exposure from overall periodontal diseases to those under treatment.

Table 1.

Under treatment 　 192 3.3 　 172 9.3 　 92 7.1 　 32 7.3 　 88 8.4 　 576 5.5 

Results

Lines 210–212

After changing the outcome from the overall prevalence of periodontal disease to that under treatment…

Discussion

Lines 255–257

(1.29 [1.03–1.62] vs. 1.43 [1.08–1.88] for overall periodontal disease; 1.77 [1.27–2.46] vs. 1.82 [1.26–2.63] for periodontal disease under treatment).

In addition to the above comments, we have re-numbered the references following the insertion of new references.

---

## [Decision Letter · Decision Letter 1]

22 Feb 2021

PONE-D-20-24726R1

Combustible cigarettes, heated tobacco products, combined product use, and periodontal disease: A cross-sectional JASTIS study

PLOS ONE

Dear Dr. Tabuchi,

Thank you for submitting your manuscript to PLOS ONE. After careful consideration, we feel that it has merit but does not fully meet PLOS ONE’s publication criteria as it currently stands. Therefore, we invite you to submit a revised version of the manuscript that addresses the points raised during the review process.

In addition to the clarifications the reviewer requests, please pay particular attention to the open access data requirement.

We look forward to receiving your revised manuscript.

Kind regards,

Stanton A. Glantz

Academic Editor

PLOS ONE

Journal Requirements:

Reviewers' comments:

Reviewer's Responses to Questions

**Comments to the Author**

1. If the authors have adequately addressed your comments raised in a previous round of review and you feel that this manuscript is now acceptable for publication, you may indicate that here to bypass the “Comments to the Author” section, enter your conflict of interest statement in the “Confidential to Editor” section, and submit your "Accept" recommendation.

Reviewer #1: (No Response)

2. Is the manuscript technically sound, and do the data support the conclusions?

Reviewer #1: Yes

3. Has the statistical analysis been performed appropriately and rigorously? 

Reviewer #1: Yes

4. Have the authors made all data underlying the findings in their manuscript fully available?

Reviewer #1: No

5. Is the manuscript presented in an intelligible fashion and written in standard English?

Reviewer #1: Yes

6. Review Comments to the Author

Reviewer #1: The authors have very carefully and thoroughly addressed almost all of the issues from the review of the initial draft in the current revision. However, 3 items were could be addressed a bit more fully.

1. The authors addressed the item about excluding surveys inconsistent with past surveys by writing “This study included data from all respondents of the 2019 survey, except those whose responses were inconsistent with the information they had provided in the earlier surveys (2015 to 2018). On the other hand, this study excluded those whose responses contained other discrepancies. For example, we excluded surveys if the respondents chose the same answer number for all questions in a set of questions, or those who reported an amount of tobacco product use but had indicated that they had never used, or were only former users of, tobacco products.”

Choosing the “same answer number for all questions” is often called “straight-lining” while reporting an amount of tobacco use after indicating they never used would be a discrepancy (or “illogical” or “inconsistent”). Should Figure 1 be referred to here?

Also the term “artificial or unnatural response” in the Figure 1 is peculiar; for the question to detect “artificial response” do the authors mean “attention checks”?

2. The DAG diagram is very helpful to understand the authors thinking with confounders, colliders, and intermediate measures. However, the rationale for BMI relating TO (not FROM) HTP use and TO (not FROM) periodontal disease should be clarified; for example, many people use tobacco products to control their weight (lowering BMI) and periodontal disease might cause people to reduce their caloric (food) intake or cease/reduce eating healthy fresh fruits and vegetables in favor of soft processed foods because of difficulties chewing, which would have BMI as an intermediate factor rather than a confounder.

3. It is not entirely clear why the personal identifying information cannot be removed and the de-identified data placed in a repository or available from the authors upon request after the requestor meets some criteria including attestation that he or she will not attempt to re-identify de-identified data.

7. PLOS authors have the option to publish the peer review history of their article (what does this mean?). If published, this will include your full peer review and any attached files.

Reviewer #1: **Yes: **Stuart Gansky, DrPH

---

## [Author Response · Author response to Decision Letter 1]

1 Mar 2021

Comments from Reviewer #1: 

The authors have very carefully and thoroughly addressed almost all of the issues from the review of the initial draft in the current revision. However, 3 items were could be addressed a bit more fully.

Response: We thank Professor Stuart Gansky again for his insightful comments. We deeply appreciate his review. We have incorporated all the changes in the manuscript to reflect the suggestions provided. The changes have been indicated with yellow markers. We hope that the revised manuscript is now suitable for publication. 

Comment 1: 

1. The authors addressed the item about excluding surveys inconsistent with past surveys by writing “This study included data from all respondents of the 2019 survey, except those whose responses were inconsistent with the information they had provided in the earlier surveys (2015 to 2018). On the other hand, this study excluded those whose responses contained other discrepancies. For example, we excluded surveys if the respondents chose the same answer number for all questions in a set of questions, or those who reported an amount of tobacco product use but had indicated that they had never used, or were only former users of, tobacco products.”

> Choosing the “same answer number for all questions” is often called “straight-lining” while reporting an amount of tobacco use after indicating they never used would be a discrepancy (or “illogical” or “inconsistent”). Should Figure 1 be referred to here?

> Also the term “artificial or unnatural response” in the Figure 1 is peculiar; for the question to detect “artificial response” do the authors mean “attention checks”?

Response: We appreciate this comment and apologize for the use of confusing words and the inaccurate description of Figure 1. 

As the reviewer instructed, “artificial or unnatural response to alcohol and drug use (n = 134)” and “comorbidities (n = 41)” in Figure 1 is what we would like to use to mean “straight-lining” responses; and “inconsistent response to smoking status (n = 326)” indicates responses with a discrepancy. In addition, artificial response to a specific question, “Please choose the second from the bottom,” indicates just the attention checks. 

Following the author’s kind instruction, we changed the manuscript and Figure 1 as follows:

Changes: 

Methods

Lines 110–120

This study included data from all respondents of the 2019 survey except those whose responses were inconsistent with the information they had provided in the earlier surveys (2015 to 2018). On the other hand, this study excluded those whose responses were straight-lining or contained discrepancies. For example, we excluded surveys as straight-lining responses if the respondents chose the same answer number for all questions in a set of questions. We also excluded respondents as responses with discrepancies if they reported an amount of tobacco product use but had indicated that they had never used, or were only former users of, tobacco products. In addition to these exclusion criteria, we performed an attention check for respondents using the question "Please choose the second from the bottom." Using this attention check, we excluded respondents who selected responses except the second answer from the bottom. 

Fig 1. Flow diagram of the study

Comment 2: The DAG diagram is very helpful to understand the authors thinking with confounders, colliders, and intermediate measures. However, the rationale for BMI relating TO (not FROM) HTP use and TO (not FROM) periodontal disease should be clarified; for example, many people use tobacco products to control their weight (lowering BMI) and periodontal disease might cause people to reduce their caloric (food) intake or cease/reduce eating healthy fresh fruits and vegetables in favor of soft processed foods because of difficulties chewing, which would have BMI as an intermediate factor rather than a confounder.

Response: We appreciate the reviewer’s insightful comment. 

We can hypothesize that BMI is related TO tobacco use because some people use tobacco to control their weight (BMI → HTP use), and periodontal disease is related TO BMI because of difficulty chewing (Periodontal disease → BMI). Another explanation may be considered, that is, BMI might be related TO HTP use because people lose their weight after using tobacco products (HTP use → lowering BMI), and periodontal disease may be related TO lowering BMI due to chewing difficulties (Periodontal disease → lowering BMI). Taken together, BMI may be either an intermediate variable or a collider; therefore, BMI may not be included in the DAG-based model. We consider that the latter explanation, that BMI is a collider in our DAG, is suitable.

Given this discussion, we changed our DAG and all the results regarding sensitivity analyses of the DAG-based model. Even in the revised DAG-based model, the consistency of the results remains unchanged.

Changes:

Lines 179–186

Furthermore, to validate the confounding selection in the main analysis, we performed a second sensitivity analysis which describes a directed acyclic graph (DAG), and constructed multivariable models based on confounders from the DAG (i.e., age, sex, BMI, educational level, routine dental checkup, secondhand exposure to combustible cigarettes from others, secondhand exposure to heated tobacco products or e-cigarettes from others, and smoking pack-years), and confirmed the consistencies of the results.

S1 Fig. A directed acyclic graph of this study

S2 Table. Results of sensitivity analysis: multivariable modified Poisson regression analysis to estimate the prevalence ratios for overall periodontal disease and periodontal disease under treatment after adjusting for DAG-based confounders.

　 　 Overall periodontal disease 　 Periodontal disease under treatment

　 PR 95%CI PR 95%CI

Smoking status 

Never user 1.00 (Reference) 1.00 (Reference)

Former user 1.63 1.41 － 1.88 1.96 1.57 － 2.43 

Current user 

Combustible cigarette use 1.41 1.12 － 1.77 1.91 1.38 － 2.66 

HTP use 1.64 1.24 － 2.17 2.15 1.42 － 3.26 

Combined use 1.84 1.45 － 2.34 2.18 1.53 － 3.09 

Demographics 

Age 

18–24 0.62 0.46 － 0.82 0.59 0.36 － 0.98 

25–34 0.61 0.45 － 0.83 0.61 0.38 － 0.99 

35–44 1.00 (Reference) 1.00 (Reference)

45–54 1.53 1.27 － 1.85 1.34 0.99 － 1.81 

55–64 2.28 1.90 － 2.73 2.34 1.78 － 3.08 

65–74 2.00 1.64 － 2.44 1.91 1.42 － 2.56 

Sex 

Male 1.00 (Reference) 1.00 (Reference)

Female 1.12 0.99 － 1.26 1.07 0.88 － 1.29 

Marital status 

Single (Reference) (Reference)

Married 1.12 0.97 － 1.31 1.36 1.06 － 1.74 

Widowed/divorced 1.16 0.94 － 1.42 1.41 1.01 － 1.95 

30-day alcohol use 

Absent 1.00 (Reference) 1.00 (Reference)

Present 1.05 0.94 － 1.17 0.89 0.76 － 1.06 

Income (USD /year) 

≤ 9,999 1.31 1.00 － 1.73 1.68 1.12 － 2.54 

10,000–39,999 1.12 0.98 － 1.27 1.13 0.92 － 1.38 

40,000–79,999 1.00 (Reference) 1.00 (Reference)

80,000–119,999 0.81 0.69 － 0.96 0.66 0.51 － 0.86 

≥ 120,000 0.81 0.65 － 1.00 0.84 0.62 － 1.14 

Unknown 0.79 0.67 － 0.93 0.68 0.53 － 0.89 

Educational level 

Less than high school 1.41 1.08 － 1.83 1.06 0.64 － 1.77 

High school graduate 1.00 (Reference) 1.00 (Reference)

College or associates' degree 0.78 0.67 － 0.91 0.84 0.67 － 1.05 

Bachelor's degree 0.85 0.75 － 0.96 0.82 0.68 － 0.99 

Master's or doctoral degree 0.77 0.59 － 1.01 0.77 0.52 － 1.16 

Routine dental checkup 

Absent 1.00 (Reference) 1.00 (Reference)

Present 1.53 1.37 － 1.70 6.29 4.91 － 8.06 

Secondhand exposure to combustible cigarettes 

Absent 1.00 (Reference) 1.00 (Reference)

Present 1.24 1.10 － 1.40 1.08 0.90 － 1.30 

Secondhand exposure to heated tobacco products or e-cigarettes 

Absent 1.00 (Reference) 1.00 (Reference)

Present 1.08 0.96 － 1.22 1.06 0.88 － 1.27 

Smoking pack-years 

≤ 5 1.00 (Reference) 1.00 (Reference)

6–10 1.02 0.74 － 1.41 1.12 0.72 － 1.75 

11–20 0.94 0.72 － 1.23 0.67 0.43 － 1.04 

21–30 0.89 0.68 － 1.17 0.84 0.56 － 1.28 

31–40 1.03 0.78 － 1.36 0.91 0.59 － 1.38 

41–50 1.23 0.91 － 1.66 1.47 0.99 － 2.20 

≥ 51 1.10 0.80 － 1.50 1.03 0.64 － 1.67 

Note. DAG, directed acyclic graph; PR, prevalence ratio; CI, confidence interval; HTP, heated tobacco products; e-cigarette, electronic cigarette; BMI, body mass index

S3 Table. Variance inflation factors to evaluate multicollinearity of each variable

Variable VIF

　 Main analysis 　 Sensitivity analysis

Smoking status 　 　

Former user 1.33 1.32

Current user 

Combustible cigarette use 2.26 2.2

HTP use 1.39 1.35

Combined use 2.43 2.23

Age 

18-24 2.21 2.2

25-34 1.56 1.56

45-54 1.81 1.8

55-64 1.97 1.89

65-74 1.94 1.81

Sex 

Female 1.43 1.36

BMI 

≤ 18.4 1.09 

25.0-29.9 1.12 

≥ 30.0 1.07 

Marital status 

Married 1.87 1.86

Widowed/divorced 1.34 1.34

30-day alcohol use 

Present 1.19 1.18

Income (USD /year) 

≤ 9,999 1.15 1.15

10,000-39,999 1.44 1.43

80,000-119,999 1.28 1.28

 ≥ 120,000 1.17 1.17

Unknown 1.43 1.43

Educational level 

Less than high school 1.1 1.09

College or associate’s degree 1.42 1.42

Bachelor's degree 1.52 1.52

Master's or doctoral degree 1.19 1.19

Routine dental checkup 

Present 1.07 1.07

Use of other tobacco products 

Present 1.18 

Secondhand exposure to combustible cigarettes from others 

Present 1.27 1.27

Secondhand exposure to HTPs or e-cigarettes from others 

Present 1.33 1.32

Smoking pack-years 

6–10 1.36 1.36

11–20 1.66 1.64

21–30 1.58 1.57

31–40 1.4 1.4

41–50 1.25 1.24

≥ 51 1.3 1.3

Comorbidities 

History of hypertension 

Present 1.26 

History of diabetes mellitus 

Present 1.13 

History of bronchitis or pneumonia 

Present 1.04 

History of heart diseases 

Present 1.09 

History of stroke 

Present 1.06 

Mean VIF 1.42 　 1.48

Note. VIF, variance inflation factors; HTP, heated tobacco products; BMI, body mass index; e-cigarettes, electronic cigarettes;

Comment 3: It is not entirely clear why the personal identifying information cannot be removed and the de-identified data placed in a repository or available from the authors upon request after the requestor meets some criteria including attestation that he or she will not attempt to re-identify de-identified data.

Response: We apologize for our unclear explanation about data availability. In addition to the restriction of ethical guidelines in Japan, our data are available only via our collaborative study framework, as explained in our study profile paper [ref]. Also, we obtained web-based informed consent that the individual data will be used only in the context of our study project and not be used in a public repository. Taken together, we cannot place our data in a public repository.

Ref (cited in Reference 13 in the manuscript)

Tabuchi T, Shinozaki T, Kunugita N, Nakamura M, Tsuji I. Study profile: the Japan “Society and New Tobacco” Internet Survey (JASTIS): a longitudinal internet cohort study of heat-not-burn tobacco products, electronic cigarettes, and conventional tobacco products in Japan. J Epidemiol. 2019;29(11): 444-450. doi: 10.2188/jea.je20180116

---

## [Decision Letter · Decision Letter 2]

10 Mar 2021

Combustible cigarettes, heated tobacco products, combined product use, and periodontal disease: A cross-sectional JASTIS study

PONE-D-20-24726R2

Dear Dr. Tabuchi,

We’re pleased to inform you that your manuscript has been judged scientifically suitable for publication and will be formally accepted for publication once it meets all outstanding technical requirements.

Kind regards,

Stanton A. Glantz

Academic Editor

PLOS ONE

Additional Editor Comments (optional):

Reviewers' comments:

Reviewer's Responses to Questions

**Comments to the Author**

1. If the authors have adequately addressed your comments raised in a previous round of review and you feel that this manuscript is now acceptable for publication, you may indicate that here to bypass the “Comments to the Author” section, enter your conflict of interest statement in the “Confidential to Editor” section, and submit your "Accept" recommendation.

Reviewer #1: All comments have been addressed

2. Is the manuscript technically sound, and do the data support the conclusions?

Reviewer #1: Yes

3. Has the statistical analysis been performed appropriately and rigorously? 

Reviewer #1: Yes

4. Have the authors made all data underlying the findings in their manuscript fully available?

Reviewer #1: Yes

5. Is the manuscript presented in an intelligible fashion and written in standard English?

Reviewer #1: Yes

6. Review Comments to the Author

Reviewer #1: The authors have thoroughly, carefully, and completely addressed the previous reviews having made revisions to the manuscript.

7. PLOS authors have the option to publish the peer review history of their article (what does this mean?). If published, this will include your full peer review and any attached files.

Reviewer #1: **Yes: **Stuart A. Gansky

---

## [Editor Report · Acceptance letter]

22 Mar 2021

PONE-D-20-24726R2 

Combustible cigarettes, heated tobacco products, combined product use, and periodontal disease: A cross-sectional JASTIS study 

Dear Dr. Tabuchi:

I'm pleased to inform you that your manuscript has been deemed suitable for publication in PLOS ONE. Congratulations! Your manuscript is now with our production department. 

Kind regards, 

on behalf of

Professor Stanton A. Glantz 

Academic Editor

PLOS ONE